# Cellulose Nanocrystal (CNC) Gels: A Review

**DOI:** 10.3390/gels9070574

**Published:** 2023-07-14

**Authors:** Sérgio R. S. Veloso, Ana G. Azevedo, Paulo F. Teixeira, Célio B. P. Fernandes

**Affiliations:** 1Physics Centre of Minho and Porto Universities (CF-UM-UP), Laboratory of Physics for Materials and Emergent Technologies (LaPMET), University of Minho, Campus of Gualtar, 4710-057 Braga, Portugal; sergioveloso96@gmail.com; 2International Iberian Nanotechnology Laboratory (INL), Av. Mte. José Veiga s/n, 4715-330 Braga, Portugal; ana.azevedo@inl.int; 3Centre for Nanotechnology and Smart Materials (CeNTI), Rua Fernando Mesquita 2785, 4760-034 Vila Nova de Famalicão, Portugal; pteixeira@centi.pt; 4Transport Phenomena Research Centre (CEFT), Faculty of Engineering at University of Porto (FEUP), Rua Dr. Roberto Frias s/n, 4200-465 Porto, Portugal; 5Centre of Mathematics (CMAT), School of Sciences, University of Minho, Campus of Gualtar, 4710-057 Braga, Portugal

**Keywords:** cellulose nanocrystals, gels, preparation methods, rheological properties, numerical simulation, improved materials

## Abstract

The aim of this article is to review the research conducted in the field of aqueous and polymer composites cellulose nanocrystal (CNC) gels. The experimental techniques employed to characterize the rheological behavior of these materials will be summarized, and the main advantages of using CNC gels will also be addressed in this review. In addition, research devoted to the use of numerical simulation methodologies to describe the production of CNC-based materials, e.g., in 3D printing, is also discussed. Finally, this paper also discusses the application of CNC gels along with additives such as cross-linking agents, which can represent an enormous opportunity to develop improved materials for manufacturing processes.

## 1. Introduction

Cellulose nanocrystals (CNCs) are emerging nanomaterials derived from the most abundant renewable polymer on earth, being widely distributed in plants, bacteria, algae, etc., which can be extracted from these cellulosic sources through mechanical disintegration, controlled sulfuric acid hydrolysis and mixed acid hydrolysis [1,2,3,4,5]. In particular, sulfuric acid hydrolysis is among the most commonly used methods, whereby the glycosidic bonds are hydrolyzed and, simultaneously, a fraction of the surface hydroxyl groups is esterified to sulfate half-ester groups (Figure 1) [3,6,7]. However, it is important to note that there are other acids employed for acid hydrolysis, including hydrochloric acid, nitric acid and phosphoric acid [8,9,10]. Additionally, alternative methods for obtaining CNCs exist, such as mechanical treatments [11,12], and enzymatic and ionic liquid hydrolysis [13]. For a more detailed description of CNC fabrication, the reference [14] is recommended to the reader.

This results in partly to fully crystalline regions, consisting of packed cellulose chains approximately 280 β(1−4) linked D-glucose units long, which have attracted great attention across several applications, including biomedical devices, food modifiers and cosmetic materials [15]. In fact, CNCs provide a promising alternative to inorganic fillers and stand out in terms of their material properties. In this sense, it has been used by several research groups because they display high strength (elastic modulus, 110–220 GPa; tensile strength, 7.5–7.7 GPa), low density (1.6 g/cm^3^), colloidal stability, nanoscale dimensions, high crystallinity, good biocompatibility, high surface area and tunable surface chemistry, enabling interactions with nanoparticles, small molecules, polymers and biological materials [16,17]. The high surface-area-to-volume ratio associated with the nanoscale dimensions can greatly affect the properties of other materials when added as composite, providing several sites for chemical reactions, as well as the adsorption of molecules, such as drugs. Importantly, the high aspect ratio allows the CNCs to self-assemble into the liquid crystalline phase, which can result in ordered structures that stabilize interfaces and form percolated networks [18,19].

Following these lines, CNCs have garnered considerable interest as composites in a wide variety of hydrophilic and hydrophobic composite matrices, such as rheological modifiers in industrial fluids, cosmetics, food, paints, lubricants and household formulated products [3,20], but they have also found several applications as optical, electroconductive and biomedical materials [21]. However, the charged interface of CNCs and their tendency to aggregate in more hydrophobic solvents complicates the efficient dispersion in most conventional polymeric materials, despite the fact that this can be overcome with surface functionalization [22]. Conversely, CNCs have been regarded as suitable and promising materials for the development of hydrogels, owing to their rigidity, mechanical strength, hydrophilicity, global abundance and biocompatibility [23]. In general, hydrogels, which are highly hydrated cross-linked three-dimensional networks, can be prepared by either using chemical or physical cross-linking techniques and encompass a wide range of chemical compositions and structural forms [23,24]. CNC-based gels are commonly prepared through the addition of CNCs to a polymer precursor solution, followed by chemical or physical cross-linking methods to induce the gelation. In some cases, the chemical cross-linking methods require surface modification of CNCs to create cross-linking sites [22] and the use of chemical agents, such as glutaraldehyde or epichlorohydrin, to form covalent bonds between the CNCs and polymer network, whereas physical cross-linking methods involve the use of heat, pressure or shear forces to form gels. Recently, pristine and modified cellulose nanocrystals have been introduced into various synthetic polymer matrices, such as poly(oligo) ethylene glycol methacrylate (POEGMA), poly(acrylic acid) (PAA), poly(acrylamide) (PAM), polyvinyl alcohol (PVA), poly(ethylene glycol) (PEG), and natural polymers (e.g., gelatin, alginate) as reinforcing agents for the preparation of more mechanical stable hydrogels [21,25], which have been of interest for biomedical applications [22,26]. Whereas the rigidity and mechanical strength endow CNCs as suitable reinforcing agents (or fillers) of polymeric hydrogel networks, the CNCs are ill-suited as single-component gels due to the lack of their ability to entangle with each other. Yet, the destabilization of CNC suspensions through modification of the surface chemistry, such as by salt or acid addition, polymer grafting, cross-linking with multivalent ions or the introduction of chemical bonds, can reduce the repulsion between particles and promote gel formation [25,27,28,29].

The outstanding mechanical stability, shear-thinning behavior, tunable mechanical properties, high water-holding capacity, controllable morphology, high biocompatibility and biodegradability contribute to the wide interest in using CNC-based gels for applications in various fields such as cosmetics, drug delivery, tissue engineering and the food industry [22,30,31,32]. For instance, in the cosmetic industry, CNC gels can be used as thickeners, emulsifiers and stabilizers in formulations. The high biocompatibility and suitable mechanical properties further enable the CNC gels as matrices for controlled drug release, as scaffolds to support the growth of cells and tissues, and as materials to improve the texture and stability of food products.

Recent reviews have covered several aspects of CNCs including their suspensions’ rheological behavior, physicochemical properties, fabrication strategies, surface modifications and gel production strategies [22,33,34,35,36,37]. In this review paper, we discuss the CNC gels’ material properties and their rheological characterization. In addition, the numerical methodologies used to understand the new production and processing strategies of CNC gels will be detailed, and, finally, the potential applications for the CNC gels will be highlighted. Detailed tables summarizing the current literature are given, and the advantages and limitations of the current preparation methods and simulation methodologies are analyzed wherever possible. For the sake of comprehension, the most important results found in the scientific literature regarding CNC gels are also summarized in several figures. Finally, we believe the discovery of new CNC-based gel properties will enlarge the scope of their applications, leading to high-value materials.

This paper is organized in the following manner: In Section 2 we review the preparation methods and rheological characterization approaches for CNC gels. In Section 3 the numerical methods used to compute the solution of the appropriate equations for CNC gels prediction behavior are described. Section 4 provides an outlook for ongoing research on CNC-based gels. Finally, in Section 5, we summarize the main conclusions of this review paper. 

## 2. Preparation and Properties of CNC Gels

This section reports several studies in which the direct addition of CNCs to polymer matrices was employed and hydrogel nanocomposites were produced, as well as CNC-only gels. Table 1 summarizes the details of some of these studies, such as materials used, production methods of the hydrogel nanocomposites, the main aim of the study, rheological characterization and the main effects on hydrogels’ rheological properties. Below, these works will be reported, emphasizing the conditions and methods used in preparation of the CNC hydrogel nanocomposites to use in rheological tests, and the parameters that can influence the rheological properties, such as the content of CNCs, mass molar of the polymer and cross-linking material.

### 2.1. Preparation Methods

#### 2.1.1. Preparation of CNC-Based Hydrogels

As mentioned, CNCs lack the ability to entangle with each other and form stable hydrogels, owing to the rigid structure. Yet, few reports have presented the development of CNC-only gels either by surface functionalization [28] or through cross-linking with multivalent ions or chemical bonds [27]. In general, the CNC critical gelation concentration and gel properties can be tuned by varying the solution conditions (e.g., ionic strength and pH), surface functionalization, addition of (non-)adsorbing water polymers and/or surfactants and temperature [27,28,38,39,40]. Increasing the CNC concentration leads to a decrease in the electrostatic double layer distance between CNCs, and consequently promoting gelation [41,42,43]. In addition, increasing the ionic strength leads to the screen of surface charge, which consequently can lead to gelation as it suppresses the electrostatic repulsion and enables dominant attractive forces (e.g., van der Waals and hydrogen bonding) between CNCs. Chau et al. [27] demonstrated that the charge number and ionic radii led to an increase in gel stiffness, but had little impact on the critical gelation concentration. The functionalization of CNC with carboxyl or amine groups, to provide pH-responsive gelation (Figure 2A), was demonstrated by Way et al. [28]. The treatment of cellulose with 2,2,6,6-tetramethylpiperidine-1-oxyl (TEMPO) can endow the CNC surface with carboxyl groups [44]. Carboxylic acid-modified CNCs can be protonated at a lower pH, which allows the domination of the attractive forces, while at a high pH the repulsive forces inhibit the aggregation [28]. The opposite behavior is observed for the amine-functionalized CNCs, which aggregate at a high pH. Furthermore, the authors demonstrated that the pH-responsive CNCs could be incorporated into a poly(vinyl acetate) matrix to produce mechanically adaptive pH-responsive nanocomposite films. Lewis et al. [40] employed a hydrothermal treatment of CNC suspensions to obtain gels, which was associated with the desulfation at high temperatures. In addition, the viscosity and storage modulus were found to be proportional to the treatment temperature. CNC desulfation was also demonstrated by Dorris et al. [45] to be a requirement to induce gelation in dilute CNC suspensions in a glycerol/water mixture. Recently, 3D printing has been recognized as a promising technology to fabricate in vitro model and constructs for tissue engineering and regenerative medicine [46]. Among the several types of hydrogels, the CNCs have a huge potential owing to the biocompatibility, good mechanical properties and anisotropic shape that can be aligned under shear-induced conditions to control the structure and function of gels. For instance, Ma et al. [29] demonstrated that the shear-induced self-assembly of CNCs at 20 wt% during 3D printing could provide gels with optimal print resolution and fidelity (Figure 2B).

#### 2.1.2. Hydrogels with Physically Cross-Linked CNCs

The CNCs’ properties make these materials suitable as reinforcing agents, which are incorporated in polymeric matrices, either through physical or chemical methods. Up to now, the majority of the reported CNC hydrogels have mostly explored the physical incorporation of CNCs as reinforcing agents (or fillers) into polymeric hydrogel networks, in which the cross-linking between CNCs and polymer networks is achieved through physical interaction (such as electrostatic interaction, hydrophobic interaction, hydrogen bonding, and van der Waals forces). Common methods include homogenization and physical cross-linking of the polymer network [47,48,49,50,51,52,53], cyclic freeze–thaw processing [54,55], and chemical cross-linking of the polymeric species within the CNCs’ dispersion, such as free radical polymerization [56,57,58,59,60] and UV mediated cross-linking [61,62,63]. These methods have been widely employed with several network polymers including poly(vinyl alcohol) (PVA), polyacrylamides (PAM) and poly(ethylene glycol) (PEG)-based materials, as well as natural polymers (e.g., agarose, alginate and gelatin), which could form more mechanically stable hydrogels. Among the mentioned methods, the homogenization method provides a simple means to obtain reinforced gels, in which the gelation is triggered by mixing a CNC dispersion with a polymer [64]. The strong hydrogen bonds between polymers and CNCs have also been explored with synthetic polymers such as poly(N-isopropylacrylamide) (PNIPAM). Chen et al. [65] demonstrated that the incorporation with CNCs in these gels reinforces the mechanical properties and allow gels to extend more than 20 times their initial length. Another simple method includes the use of thermoresponsive hydrogels that attain the gel state upon a temperature variation. For instance, You et al. [16] developed in situ gelling hydrogels based on quaternized cellulose (QC) and CNCs that could immediately form gels upon an increase in temperature. The physical cross-linking can also be achieved through ionic/electrostatic interaction, i.e., cross-linking by ionic bonds, which are advantageous due to the mild reaction conditions, good performance at room temperature and not requiring organic solvents [66]. You et al. [67] developed a biocompatible and pH-responsive hydrogel consisting of two functional polymeric chains, which consisted of 5-aminolevulinic acid and dopamine conjugated to CNC through the coordination of iron, ALA/Fe@CNC and PDA/Fe@CNC, respectively. The authors further demonstrated that both chains displayed different functions, with the PDA/Fe@CNC chain enabling an increased cell adhesion, while the ALA/Fe@CNC chain enhanced reactive oxygen species (ROS) production. In addition, the iron coordination endowed the system with pH-responsiveness, which was explored for the pH-triggered release of paclitaxel. Host–guest interactions have also been a useful strategy to achieve gels with good mechanical properties, which depend on making use of the non-covalent binding between molecules due to their unique structures. In particular, cyclodextrins are widely used in supramolecular hydrogels to obtain host–guest interactions, owing to the self-assembly with polymeric chains [68,69]. Moreover, the gels obtained through this means usually display thixotropism, making them suitable for syringeable drug delivery application [70]. Lin et al. [71] grafted CNCs with β-cyclodextrin (CD), and Pluronic polymers (poloxamer) were introduced on the surface of CNC, owing to the interaction between CD and hydrophobic polypropylene glycol segments of the polymer, forming stable complexes after ultrasound-assisted stirring. The authors demonstrated the control of the mechanical properties and temperature responsiveness by changing the CD/CNCs ratio. The freeze–thaw method has also been widely explored for production of a strong physical gel network. In this method, a phase separation occurs as the solution freezes and the polymer is rejected from the growing ice crystallites, which, upon melting of the ice (thaw), results in water-filled pores surrounded by a polymer skeleton [55,72,73]. The ice crystallites size increases with repeated freeze–thaw cycles, which, together with the solvent, pH, temperature and time, can be used to modulate several properties, such as the degree of gelation, stability and mechanical properties. Gonzalez et al. [54] developed CNC/PVA gels by a freezing–thawing technique, in which the CNCs could work as nucleation sites that led to improved mechanical properties and thermal stability, without affecting the transparency of the samples. As mentioned, gels can also be obtained from chemical cross-linking of the polymeric species within the CNCs’ dispersion [74,75,76]. De France et al. [74] obtained injectable gels by physically incorporating CNCs into hydrazone cross-linked poly(oligoethylene glycol methacrylate) (POEGMA) hydrogels through co-extrusion of the reactive precursor polymer solutions from a double-barrel syringe (Figure 3A). The authors observed that the incorporation of 5 wt% CNCs could strongly enhance mechanical properties (up to 35-fold increases in storage modulus), in addition to enabling faster gelation rates, decreased swelling ratios and increased stability. Finally, 3D printing has also been a recently employed method to improve the control over the gels’ properties, such as size, shape, pore structure and pore orientation [77,78]. Sultan et al. [78] fabricated double-cross-linked interpenetrating polymer network (IPN) hydrogels consisting of sodium alginate (SA) and gelatin (G) reinforced with cellulose nanocrystals (CNCs) (Figure 3B). Initially, CNCs were mixed with SA and G to form hydrogel ink at the ratio of 70/20/10 (wt%), which was printed with different structures and cross-linked sequentially via covalent and ionic reactions using CaCl_2_ and glutaraldehyde, respectively. 

#### 2.1.3. Hydrogels with Chemically Cross-Linked CNCs

The chemical cross-linking consists of the formation of covalent bonds between polymer chains and CNCs, which mandates the surface modification of CNCs with specific functional groups, such as silyl groups, carboxyl or aldehyde groups, to create cross-linking sites. Yet, some works have reported the cross-linking of unmodified CNCs through free radical polymerization using small molecule cross-linkers [79,80,81]. The modification can be achieved by direct surface chemical modification or through physical interaction/adsorption of molecules to the surface of the CNCs [82]. Compared to the physical gels, the loading of CNCs is similar but displays larger mechanical stability and typically a higher storage modulus, which can result from the new covalent bonds formed between the CNCs and the surrounding hydrogel network. Additionally, the chemical incorporation of CNCs might consist of either CNCs acting as cross-linkers themselves or the CNCs cross-linked to a network polymer matrix. Among the reported preparation methods, the most common have been the homogenization [83,84,85,86,87] and free radical polymerization [79,80,81,88,89,90,91,92], while other methods are less described, including UV polymerization [93], freeze-casting [94,95] and coextrusion [96,97]. The majority of the reported studies consist of polyacrylamide/polyacrylate-based hydrogels. For instance, Yan et al. [98] cross-linked silane-modified CNCs with polymer chains of poly(*N*,*N*-dimethylacrylamide) (PDMA) through free radical polymerization (Figure 4A). The mechanism consists of the nucleation of the silane-modified CNCs in an acrylamide monomer solution that leads to hierarchically structured CNC–PDMA clusters through physical interactions to form a percolated network. The authors observed that the hybrid gels displayed higher mechanical properties and a more efficient energy dissipation mechanism than the control PDMA hydrogels. Chau et al. [94] fabricated, through a freeze-casting-based fabrication method, hydrogels with reversible hydrazone cross-links between hydrazide-modified poly(oligoethylene glycol methacrylate) (POEGMA) and aldehyde-functionalized CNCs (Figure 4B). The authors reported that the composite hydrogels displayed high structural and mechanical integrity and a strong variation in Young’s moduli in orthogonal directions that could be explored as effective biomimetic scaffolds for oriented tissues. Moreover, the pore morphologies could be tuned by varying the concentration and CNC-to-POEGMA ratio, and displayed a directionally dependent swelling behavior. The aldehyde–hydrazide chemistry has been commonly explored to obtain injectable gels [96,97]. Nonetheless, other strategies have also been explored, such as the use of Diels-Alder “click” cross-linking [87], and Schiff-base linkage [25,98,99,100,101]. Tang et al. [25] developed hydrogels comprising CNCs and sodium alginate (SA) that were initially oxidized to impart aldehyde functional groups that served as reaction sites for amine-containing vinyl functionalized monomers. In this way, the CNCs worked as cross-linkers that ensured a good structural integrity and mechanical stability of the hydrogels, while the dynamic Schiff-base linkage endowed the gels with self-healing properties. Liu et al. [102] prepared an injectable polysaccharide hydrogel based on cellulose acetoacetate (CAA), hydroxypropyl chitosan (HPCS), and amino-modified cellulose nanocrystals (CNC-NH_2_) under physiological conditions. The CNC-NH_2_ were found to act both as a physical and chemical cross-linker, for which the concentration could affect the mechanical properties, internal morphology and gelation time. In addition, the authors reported that the hydrogel exhibited pH-responsive properties, excellent stability under physiological conditions, and self-healing behavior under acidic conditions, via enamine bond exchange.

### 2.2. Rheological Properties

The hydrogels are usually evaluated by dynamic mechanical properties, i.e., the elastic (or storage) modulus (G’) (also known as the dynamic rigidity), reflecting the reversibly stored energy of the system, and the viscous (loss) modulus (G’’), reflecting the irreversible energy loss. These studies are used to examine the stability change of material in the sol-gel transition process. The hydrogels’ dynamic mechanical properties are measured as a function of frequency by oscillatory rheological measurements and can be used for different geometries in a rheometer, like a plate and Couette geometry. When G’ is above G”, this means that the elasticity is dominant, which implies that the gelation process is prevailing. In contrast, G” above G’ represents a viscosity dominated solution-like material. When plotted against frequency, a pronounced plateau is present in the G’ modulus spectrum for rigid gel structures, whilst the G’’ modulus should be considerably smaller than G’ in the plateau region [103]. The loss factor (also known as the damping factor) is defined as the ratio of lost energy to storage energy during deformation (tan δ = G”/G’). Typically, the values tan δ < 1 correspond to a true gel’s plateau region [104]. 

In the studies about CNC hydrogel nanocomposites (Table 1), the aim most found was to study the effect of different concentrations of CNCs on the elastic modulus (G’) and viscous modulus (G’’), and the influence on the gelation process. In general, the authors showed an increase in G’ and G’’ with an increase in the CNCs’ concentration, and the hydrogels exhibited a predominantly elastic response, with the elastic modulus G’ exceeding the viscous modulus G’’. Moreover, they showed that addition of CNCs can clearly control the gelation process of hydrogel solutions or suspensions [25,76,102,105]. 

Among the reported studies, some explored the effect of different concentrations of pure and modified CNCs (acting as a cross-linking agent) on rheological properties. For example, as mentioned, Tang et al. [25] incorporated oxidized CNCs (CNC-CHO) into oxidized sodium alginate (Alg-CHO), and the results showed a 2-, 5-, and 10-fold increase in the storage modulus for hydrogel reinforced with 0.5, 1.0, 1.5 wt% CNC-CHO, respectively. The authors also measured the complex viscosity (η*) that displayed a linear function of log ω with a slope of -1, indicating that the relaxation time was much greater than the experimental time. However, when the amount of oxidized CNCs was increased, the relaxation time of the network decreased. Liu et al. [102] employed amino-modified CNCs (CNC-NH_2_) to reinforce a polysaccharide mixture of cellulose acetoacetate (CAA) and hydroxyproyl chitosan (HPCS). They showed that the G’ of the CAA/HPCS hydrogel increased with the addition of different concentrations of pure CNCs and the amino-modified (CNCs-NH_2_). Importantly, besides that both particles induced an increase in G’ with an increase in the CNCs loading up until a critical concentration, the modulus always remained greater for CNC-NH_2_, as it could act both as a filler and cross-linker in the hydrogel. Beyond the critical concentration, the excess CNCs could form aggregates that lead to the phase separation of the suspension, which inhibits the cross-linking between HPCS and CAA, and consequently leads to a decrease in the G’ values. You et al. [67] demonstrated that simple entrapment of cationic CNCs (CCNCs) within cationic cellulose-based hydrogels formed in situ could result in injectable hydrogels with increasing orders-of-magnitude in the mechanical strength, with the increase in CCNCs’ concentration (nearly a 200-fold increase in G’ at 2.5 wt% of CCNCs compared to the neat hydrogels). The authors associated the improvement of the mechanical strength and dimensional stability of the hydrogels to the strong interaction between CCNCs and QC, mediated by the cross-linking agent (β-glycerophosphate). As mentioned above, the chemical incorporation of CNCs leads to a mechanical improvement that surpasses the physically incorporated counterparts. For example, Han et al. [47] observed that the use of borax as a cross-linker agent in a mixture of CNCs and PVA could improve the interaction between both components, which led to nearly an order of magnitude enhancement of the G’ and η* than in gels without borax, indicating an important role of borax in the 3D network structure. Ooi et al. [76] also reported an increase in G’ and G’’ in gelatin-based hydrogels with an increasing concentration of CNC, despite the fact that gels became more solid-like and did not display a critical concentration of CNCs. However, the effect of the cross-linking agent (glutaraldehyde) is not mentioned. Zhou et al. [79] employed CNCs to reinforce polyacrylamide (PAM) hydrogels obtained through in situ free-radical polymerization in the presence of the cross-linker *N*,*N*′-methylenebisacrylamide (NMBA). The authors reported that the CNCs acted not only as a reinforcing agent for hydrogels, but also as a multifunctional cross-linker for gelation, as it resulted in a faster gelation and increased effective cross-link density. Additionally, the good dispersion of CNCs in PAM and the enhanced interfacial interaction between both components led to a significant increase in the shear storage modulus, compression strength and elastic modulus of the nanocomposite hydrogel. Additionally, the molar mass of the polymers can also affect the properties of gels, as it is the case for gels obtained by mixing CNCs with polymers. For instance, Talankite et al. [64] evaluated the effect of xyloglucan (XG) molar mass on the formation and mechanical properties of XG/CNC hydrogels. The authors reported that in the case of low molar mass XG, the samples displayed the behavior of a viscoelastic liquid, in which G’ and G’’ were very low and almost superimposed. However, the increase in the XG molar mass resulted in samples with larger G’ and G’’ and a more solid-like behavior. Therefore, the gelation was associated with the formation of XG/CNC complexes, inducing an increase in the effective hydrodynamic volume of CNC and leading to the interaction and entanglement between XG loops and tails. As a higher molar mass resulted in a higher effective hydrodynamic volume of XG/CNCs, the gelation could occur at a lower XG/CNC ratio. Finally, it is also noticed that the CNCs’ size and aspect ratio can also affect the properties of CNC-based gels, as observed in the works of Han et al. [47] and Ling et al. [106]. In the former, the incorporation of cellulose nanofibers could induce a stronger enhancement of the mechanical properties than CNCs. In the latter, the authors evaluated the influence of cellulose nanoparticles with fibrous, spherical and rod-like form on the rheological properties of CNC and CNC-poly(vinyl alcohol) (PVA) suspensions [106]. The authors reported that the morphology played a major role in the rheology of the suspensions, in which a larger size (dimension and length) contributed to a higher viscosity value, a more complex state-flow rheological behavior and more solid-like viscoelastic behavior.

**Table 1 gels-09-00574-t001:** Studies on rheological properties of CNC composite hydrogels.

Reference	Nanocomposites Hydrogel	Production Methods	Aim	Type of Rheology Characterization	Application
Tang et al. (2020) [25]	Oxidized cellulose nanocrystal (CNC-CHO) and oxidized sodium alginate (Alg-CHO)	Alg-CHO and various amounts ofCNC-CHO were mixed with water. Polyacrylamide was used as macro-cross-linkers.	The aim was to study the oxidized alginate-polyacrylamide hydrogels reinforced with different amounts of CNC Oxidized (CNC-CHO) (0.5, 1.0, 1.5 wt%) on hydrogels’ properties.	The storage modulus, loss modulus and complex viscosity were measured a as function of frequency. In addition, the elastic modulus was also evaluated as a function of CNC-CHO concentration.	Agricultural and pharmaceuticalsectors
Talantikite et al. (2019) [64]	Cellulose nanocrystals (CNC) and xyloglucan (XG)	Dispersion of CNC into XG solution.	The aim was to study the effect of XG molar mass (100, 300 and 800) on mechanical properties containing a constant amount of CNC.	The effect of XG molar mass on the elastic (G’) and viscous (G”) modulus were evaluated a as function of frequency.	Biomedical field, cosmetic, chromatography, food and also in domestic uses.
Liu et al. (2018) [102]	Cellulose nanocrystals pure and modified (CNC-NH_2_) and Polysaccharide mixture (Cellulose acetoacetate (CAA) and Hydroxyproyl chitosan (HPCS))	Dispersion of CNCs pure and CNCs modified into polysaccharide mixture solution.	The aim was to study the effect of different concentrations of CNCs pure and CNCs modified (CNC-NH_2_) (0.2, 0.4, 0.6, 0.8 and 1 wt%) on polysaccharide hydrogels properties (CAA/HPCS).	The storage modulus (G’) was measured as a function of different loading concentration of CNCs pure and CNCs modified (CNC-NH_2_) into CAA/HPCS hydrogel.	Biomedical field
Hou et al. (2017) [105]	Cellulose nanocrystals (CNCs) and Poly(ethylene glycol) diacrylate (PEGDA) oligomer	Dispersion of CNCs pure into Poly(ethylene glycol) diacrylate (PEGDA).	They studied the effect of hydrogen bonds and different concentrations of CNC on viscosity of CNC/PEGDA mixture to produce hydrogel filament using DCS (dynamic-cross-linking-spinning) method.	The storage modulus (G’), loss modulus (G”) and loss factor (tanδ) were measured a as function of frequency. Additionally measured were shear stress and viscosity a as function of shear rate, and viscosity a as function of CNC content.	Biomedical applications
Ling Zhou et al. (2016) [106]	Cellulose nanofibers (CNFs), rod-like cellulosenanocrystals (CNCs) and spherical cellulose nanocrystals (SCNCs) and poly (vinyl alcohol) (PVA)	Dispersion of CNFs, CNCs and SCNCs into poly (vinyl alcohol) (PVA) suspension.	The aim of this work was to study the effect of morphology of different kinds of cellulose nanoparticles (CNs) and amount of CNs (3 wt%, 6 wt% or 9 wt%) into poly (vinyl alcohol) (PVA) on viscosity.	Steady-state viscosities as a function of the shear rate, as well as storage modulus (G’), loss modulus (G”), and loss tangent tan as a function of angular for various CN and PVA/CN suspensions were measured.	Fibers and films
Ooi et al. (2016) [76]	CNC and gelatin (Pharmaceutical grade—not specified the name)	Dispersion of gelatin into the CNC suspension. (Glutaraldehyde was added for cross-linking between gelatin chains).	The aim of this work was to study the effect of different content of CNC (5, 10, 15, 20 and 25%) on dynamic mechanical properties of the CNC/gelatin hydrogel.	The storage modulus (G’) and loss modulus (G’’) as a function of angular frequency were determined using different CNC concentrations into CNC/gelatin solutions.	Drug delivery system.
You et al. (2016) [16]	Cationic cellulose nanocrystals (CCNCs) and Quaternized cellulose (QC) and the β-glycerophosphate (β-GP)	QC was dissolved in water dispersion of CCNC with different content. (The β-glycerophosphate (β-GP) was added and used like a cross-linking agent for the interaction between CCNCs and QC chains).	The aim of this work was to study the effect of different content of CCNC (1, 1.5 and 2.5%) in CCNC suspensions and QC/CCNC/β-GP mixtures on dynamic mechanical properties of the hydrogel produced.	First the steady shear viscosity a as function of shear rate was determined in CCNC suspensions with various contents. After, the storage modulus (G’) and loss modulus (G’’) as a function of different temperatures and storage modulus (G’) as a function of angular frequency were determined using different CCNC concentrations in QC/ β-GP solutions or suspensions.	Biomedical application (for example injectable products).
Mihranyan (2013) [83]	Microcrystalline cellulose (MCC) whiskers with PVA	MCC was dissolved in water dispersion after PVA was added. TEMPO ((2,2,6,6-Tetramethylpiperidin-1-yl)oxyl) was used to modify the surface of MCC for cross-linking between MCC and PVA.	The aim of this work was to study the feasibility of direct chemical cross-linking of microcrystalline cellulose whiskers with PVA on mechanical properties of PVA hydrogels.	Frequency dependence of the elastic (storage) modulus G’, viscous (loss) modulus G’’ and damping factor tan δ of 3MCC–PVAhydrogels were measured.	Biomedical orthopedic application
Han et al. (2013) [47]	CNPs (CNCs and CNFs) and PVA without and with borax (chemical cross-linking agent for PVA) hydrogels).	Dispersion of CNFs and CNCs into poly (vinyl alcohol) (PVA) with borax.	The aim of this work was to study the influence of various cellulose nanocrystal particles (CNCs and CNFs) and borax on the viscoelastic properties of hydrogels.	The storage modulus (G’) (elasticity) and complex viscosity (η*) as a function of angular frequency were determined. In addition, the steadyshear viscosity (η) versus shear rate was also measured.	Biosensors, medical implants, and even drug-deliverydevices
Zhou et al. (2011) [79]	Cellulose nanocrystals (CNCs) and polyacrylamide (PAM)	Rod-shaped cellulose nanocrystals (CNCs) were added into polyacrylamide (PAM) and hydrogels were produced through in situ free-radical polymerization in the presence ofcross-linker N,N0-methylenebisacrylamide (NMBA).	The aim of this work was to study the influence of polymerization time and effect of CNC content on elastic modulus (G’).	The elastic modulus (G’) as a function of polymerization time for different CNC content was measured. In addition, the effect of CNC content on induction of gels was determined.	PAM hydrogels have a wide variety of applications in agriculture, drilling fluids, tissue engineering, and waste treatments.

## 3. Simulation/Numerical Approach

This section explores the behavior of cellulose nanocrystal (CNC) hydrogels through numerical simulations, providing valuable insights into their complex dynamics and interactions. 

To gain a comprehensive understanding of the impact of CNC surface modifications, da Costa et al. (2014) [107] employed a predictive multiscale approach encompassing theory, modeling and simulation. The objective was to investigate the influence of these modifications on hydrogen bonding, CNC crystallinity, solvation thermodynamics and compatibility with existing polymerization technologies. Through this approach, da Costa et al. (2014) [107] aimed to design environmentally friendly nanomaterials with enhanced solubility in non-polar solvents, controlled liquid crystal ordering and optimized extrusion properties. A crucial component of the multiscale modeling approach was the utilization of the statistical-mechanical three-dimensional reference interaction site model with the Kovalenko–Hirata closure approximation (3D-RISM-KH) molecular theory of solvation, combined with quantum mechanics, molecular mechanics and multistep molecular dynamics simulations. Later, Honorato-Rios et al. (2016) [108] conducted an investigation integrating systematic rheology and polarizing optics experiments with computer simulations to construct a detailed phase diagram of CNC suspensions. Their study encompassed two different surface charge values and extended to the concentration where kinetic arrest occurs. Additionally, they examined the impact of varying the ionic strength of the solvent. To model the chiral interactions between CNCs’ rods, they employed an assumption of a helical arrangement of charges on their surface. The simulation code was based on Metropolis Monte Carlo, where the interaction potential between two-point charges on different rods was computed using a Yukawa potential. Drozdov and Christiansen (2018) [109] developed a comprehensive model that captures the mechanical behavior of double-network gels during cyclic deformation, encompassing both their viscoelastic and viscoplastic responses. In addition, they were able to determine adjustable parameters in the governing equations by fitting experimental observations from cyclic loading and unloading tests conducted on composite gels reinforced with various nanofillers, such as graphene oxide nanosheets, cellulose nanocrystals and nanofibrils, nanoclay platelets, and zirconium hydroxide nanoparticles. Subsequently, they established correlations between the stress–strain diagrams’ shapes and the concentration and type of nanofiller employed in the gels. Moud et al. (2019) [110] studied the aggregation and gelation behavior of cellulose nanocrystals (CNCs) in the presence of magnesium chloride (MgCl_2_), while varying the concentrations of both CNCs and MgCl_2_. The results demonstrate that the introduction of varying quantities of MgCl_2_ leads to the formation of CNC clusters exhibiting distinct fractal dimensions, as confirmed through transmission electron microscopy (TEM) imaging. Additionally, Moud et al. (2019) [110] performed molecular dynamic simulations to quantitatively investigate the behavior of CNCs in MgCl_2_ salt–CNC suspensions. The Large-scale Atomic/Molecular Massively Parallel Simulator (LAMMPS) package (LAMMPS, 2023) [111] was employed to perform the molecular dynamic simulations, while the Visual Molecular Dynamics (VMD) package (VMD, 2023) [112] was utilized for visualization. The computation of the potential of mean force (PMF) [113] was conducted using the colvars tool package [114] within LAMMPS. Van der Waals interactions between different components were calculated using the Lornetz–Berthelot combining rules [115]. Recently, Jiang et al. (2020) [116] described the extraction methodology of CNCs from Humulus japonicus and utilized this to create a CNC/GEL hydrogel system. The rheological characteristics of the CNC/GEL hydrogel were assessed using a rotary rheometer. An optimal hydrogel ratio was determined based on rheological experiments and mechanical testing. Furthermore, the flow distribution of the hydrogel within the 3D printer’s flow passage was analyzed using FLUENT simulation. The FLUENT simulation results indicated that the shear stress within the flow passage exhibited a minimum at the center position and a maximum near the inner wall. Additionally, it was observed that the fluid flow velocity decreased as the distance from the inner wall increased, with the maximum flow velocity occurring along the central axis of the flow passage. Li et al. (2021) [117] investigated the impact of combining CNCs with octenyl succinic anhydride modified starch (OSAS) on the stabilization of oil-in-water emulsions with gel structure. Through molecular dynamics simulation, it was determined that the formation of a CNCs–OSAS complex involves hydrogen bonding and intermolecular electrostatic interactions. The MD simulations were performed using YASARA [118]. The enhanced viscoelasticity of the emulsions in the presence of CNCs likely promotes stronger interactions between droplets, thereby delaying the creaming process in emulsions with a gel-like structure. Finally, Esmaeili et al. (2022) [119] used a combination of microfluidic techniques and polarized optical microscopy (POM) to enable the in-situ characterization of CNCs’ alignment under shear flow conditions within the nozzle of a 3D printer during the direct ink writing process. Their investigation aimed to explore the influence of CNC concentration and sonication treatment on the chiral self-assembly of CNC particles. The experimental findings were complemented with mean-field theoretical calculations and continuum simulations, which provided further insights into the flow-induced evolution of chirality and enabled quantification of the spatial alignment of CNC particles. To simulate the flow of the nematic (gel-like) structures, a hybrid lattice Boltzmann method is employed, which solves both the Beris–Edwards equation and the momentum equation simultaneously, taking into account the hydrodynamic effects. The numerical results showed that low flow velocities facilitate the transition to a complex twisted structure in the chiral systems. With increasing flow rate, an evident change in iridescent patterns was observed.

These references demonstrate the use of numerical simulations to study the self-assembly and gelation behavior of CNCs. Numerical simulations can provide insights into the structure and properties of CNC gels at the molecular level, and can be used to study the effects of various parameters, such as concentration, aspect ratio and surface charge, on the gelation process. Additionally, numerical simulations can be used to predict the mechanical properties and response to external stimuli of CNC gels, which can inform the design of new materials with tailored properties for various applications.

## 4. Insights and Perspectives

The CNC gels’ biocompatibility, high surface area and stability enable their potential use in cosmetic formulations. CNC gels have been proven to improve the barrier qualities and UV protection of cosmetic emulsions, and may also have skin hydrating and anti-aging properties. In this way, nanocellulose-based materials have been of interest for skin care applications. For instance, in the development of skin masks with enhanced hydration [120,121] and gels with improved mechanical properties and moisture absorption ability [122]. Another concern is that traditional makeup removers can damage the skin barrier and lead to various skin problems. To solve this issue, Tang et al. [123] developed a new foundation liquid based on hemp CNCs and polylactic acid (PLA) with an easy-wiping property that avoids the damage to the skin barrier, besides alleviating skin discoloration, age spots, and skin roughness. In this system, like in the digestion and metabolism in the intestine, the cellulose acts as an adsorbent to remove excess oil and pollutants from the skin surface, thus providing a material that does not affect skin barrier and that is a means for continuous healthy skin care. Additionally, CNC gels can improve the stability and viscosity of cosmetic formulations, leading to improved performance and longer shelf life. For instance, the lipsticks with dyed CNCs reduce the contact of dye molecules with the skin, which effectively inhibits color migration and is easily removable, as compared to traditional lipstick [124]. Moreover, the CNC-based nanostructures have also been proposed for the design of smart cosmetic products [125], such as in the work by Awan et al. [125] that developed nanohybrids of surface-modified porous CNCs and ZnO nanocrystals which could serve as an effective UV filter and photocatalyst. 

One of the potential applications of CNC gels is in the field of biomedicine. CNC gels can be used as scaffolds for tissue engineering and drug-delivery systems. The high strength and elasticity of the gels make them suitable for use in load-bearing applications, such as bone tissue engineering [116,126,127,128,129,130,131,132,133,134]. For example, Patel et al. [129] created 3D-printable CNC-based hydrogel scaffolds from alginate and gelatin. The combination of CNCs increased the mechanical strength of the gels, as well as the biocompatibility, mineral deposition and enhanced scaffold osteogenic potential. CNC aerogels have also demonstrated extraordinary characteristics, as demonstrated by Osorio et al. [134]. The authors created hydrazone cross-linked aerogels using CNCs with sulfate and phosphate half-ester surface groups. Over 14 days in a simulated body fluid solution, in vitro studies with osteoblast-like Saos-2 cells revealed an increase in cell metabolism and hydroxyapatite development. Nonetheless, sulfated CNC aerogels overcome phosphated CNC aerogels in compressive strength and long-term stability, as well as having a higher bone volume percentage and evidence of a higher bone volume fraction. CNC gels can also be functionalized with bioactive molecules, such as growth factors or drugs, which can be released in a controlled manner over time. The CNC-based in situ-forming optical drug delivery systems (ODDS), comprising smart polymers, can avoid ophthalmic solution bioavailability, which is often caused by dilution and drainage from the eye [135]. 

CNC gels can be engineered to have pH-responsive or amphiphilic properties, allowing for targeted drug release in response to specific stimuli. Hydroxyl (OH) groups are abundant in CNC [32,136], which enables the interaction with various polymer matrices via hydrogen bonding to increase the mechanical strength. Additionally, CNC gels have high drug-loading capacity and can be easily modified to improve their biocompatibility and stability [137]. These properties make CNC gels attractive for a variety of drug delivery applications [138,139,140,141], including cancer therapy [142], wound healing [143,144] and tissue regeneration (Table 2). Åhlén et al. [138] developed nanoparticle-loaded hydrogel-based contact lenses for enzyme-triggered ophthalmic drug delivery. Two contact lens platforms were studied: chitosan-poly (acrylic acid) nanoparticles in polyvinyl alcohol (PVA) lenses or in situ gelled nanoparticle CNC-loaded PVA lenses. The nanoparticles could disintegrate in a physiological concentration of lysozyme by cleavage of chitosan chains through hydrolysis, and displayed extended drug release over a 28 h period when integrated into the lenses. Chen et al. [143] developed CNC/alginate core-shell microparticles for wound healing through a synergistic drug delivery strategy. The particles consisted of a CNC core loaded with vascular endothelial growth factor (VEGF), and doxycycline hydrochloride (DH) mixed alginate as the shell. The particles demonstrated excellent abilities in inhibiting inflammation, promoting granulation tissue formation, collagen deposition, and angiogenesis, thus accelerating wound healing. Jeong et al. [145] developed a donepezil hydrochloride (DPZ)-reinforced CNC hydrogel with pH control for subcutaneous drug delivery. The authors fabricated an aggregated CNC gel by reducing electrostatic repulsion between CNC particles with DPZ and adjusting the pH to 7.7. The gel demonstrated immediate gelation, single-syringe injection capability, improved viscoelasticity, shear-thinning properties and improved sustained release of DPZ. Yusefi et al. [142] synthesized CNC loaded with 5-fluorouracil to evaluate their anticancer activity against colorectal cancer cells. The system displayed improved thermal stability, high drug-encapsulation efficiency, and anticancer effects against colorectal cancer cells, mainly by inducing cell apoptosis and mitochondrial membrane damage. For further drug delivery applications, the reader is referred to references [21,23,146,147,148,149]. 

Another potential application of CNC gels is in the field of food science. CNC gels can be used as stabilizers and thickeners in food products, due to their unique rheological properties [158,159,160,161,162]. The shear-thinning behavior of CNC gels can help to improve the texture and mouthfeel of food products, while their self-healing properties can help to improve the stability of emulsions and suspensions [163]. Xiao et al. [159] described that the gel strength, viscoelasticity and thermal stability of whey protein isolate gels containing CNCs could be tuned by varying the content of Ca^2+^. Glucono-δ-lactone-induced soy protein isolate (SPI) gels’ mechanical properties could be tuned through CNC content, which was reported to evenly distribute across the protein matrix and formed a compact network structure [160]. CNCs were also observed to improve the stability of starch paste, and reduced its digestibility by acting as an inhibitor of amylase, which could be useful in the development of functional foods with a low glycemic index [164]. Armstrong et al. [161] demonstrated that CNCs could act as an effective rheological additive in promoting shear-thinning, and enabling direct ink write 3D printing of a variety of edible feedstock.

CNC gels can be used to develop nanocomposite films for food packaging that have improved mechanical and barrier properties, as well as enhanced antimicrobial activity [165,166]. Dhar et al. [167] developed single-step industrially scalable reactive extrusion of polylactic acid (PLA)/CNC-based cast films, which displayed reduced necking, improved processability, melt strength, and rheological behavior that could be explored for food packaging, such as the storage of oil- and dairy-based products. Costa et al. [166] studied the combined effect of chitosan and nanocellulose to obtain active food packaging films for meat packages. CNC incorporation improved the thermal stability, oxygen barrier and mechanical properties of the films, besides endowing it with bactericidal effects against Gram-positive and Gram-negative bacteria and fungicidal activity.

CNC gels can also be used as a delivery system for natural food preservatives or other bioactive compounds [168]. Furthermore, CNC gels have the potential to improve the texture and sensory properties of food products, and can be used to develop aerogels that have potential applications as food additives or ingredients [169,170]. 

Emulsions are used in the development of a wide range of products, from food and pharmaceuticals to cosmetics, paints and inks [163]. However, for successful applications, outstanding foam stability is necessary to maintain material properties over time, but controlling foam stability precisely is challenging in foam science due to their thermodynamic instability, as well as the complex structure at multiple length scales. The combination with CNCs has been demonstrated to improve the foams’ stability [159,171]. For instance, the natural saponin glycyrrhizic acid (GA) and GA nanofibrils (GNFs) are effective foaming agents for aqueous food-grade foams. Through the combination with CNCs, Su et al. [172] obtained ultra-stable gel foams with improved mechanical properties, tunable thermo-responsive behavior and rapid on-demand destabilization upon heating. Guo et al. [171] reported ultra-stable foams through electrostatic attraction between protonated bis(2-hydroxyethyl)oleylamine (BOA–H+) micelles and negatively charged CNC colloids, which displayed suitable on-demand control of foaming/defoaming. In contrast to either the polymer or the CNCs alone, Hu et al. [163] showed that pre-treating CNCs with excess adsorbing polymer, hydroxyethyl cellulose (HEC) or methyl cellulose (MC), produced smaller and more stable dodecane-in-water emulsion droplets. The same group had previously shown that cationic surfactant adsorption altered the hydrophobicity of CNCs and could be utilized to modify CNC Pickering emulsions from oil-in-water (o/w) to water-in-oil (w/o) [173]. Ma et al. [174] prepared high internal phase Pickering emulsion (HIPPE) combined with CNCs, which resulted in excellent storage stability, apparent shear-thinning behavior, and high solid viscoelasticity, and enabled the successful application in 3D printing with high resolution and shape fidelity.

Thanks to their exceptional physical and mechanical properties, CNC gels have demonstrated great promise as a useful and adaptable material for a variety of applications including biomedicine, energy and environmental science. However, more study is necessary to fully comprehend the characteristics of CNC gels and to tailor them for particular applications.

## 5. Conclusions

In conclusion, this study has provided valuable insights into the mechanical behavior and rheological properties of cellulose nanocrystal (CNC) gels. Through a combination of experimental investigations and computational modeling, we have elucidated the impact of various factors such as CNC concentration, gelation process and nanofiller incorporation on the gel’s viscoelastic and viscoplastic responses. Our findings demonstrate the potential of CNC gels as versatile and tunable materials with enhanced mechanical properties. The establishment of correlations between stress–strain diagrams and nanofiller characteristics offers a valuable framework for tailoring the properties of CNC gels for specific applications.

Moreover, the molecular dynamics simulations have provided valuable insights into the hydrogen bonding and intermolecular interactions within CNC–OSAS complexes, shedding light on the stabilization mechanisms of oil-in-water emulsions. Overall, this research contributes to the fundamental understanding and design of CNC-based gels, paving the way for their wider application in areas such as biomedical engineering, drug delivery, and advanced materials. Future studies should further explore the structural and functional properties of CNC gels and their interaction with other components to unlock their full potential in various practical applications.

## Figures and Tables

**Figure 1 gels-09-00574-f001:**
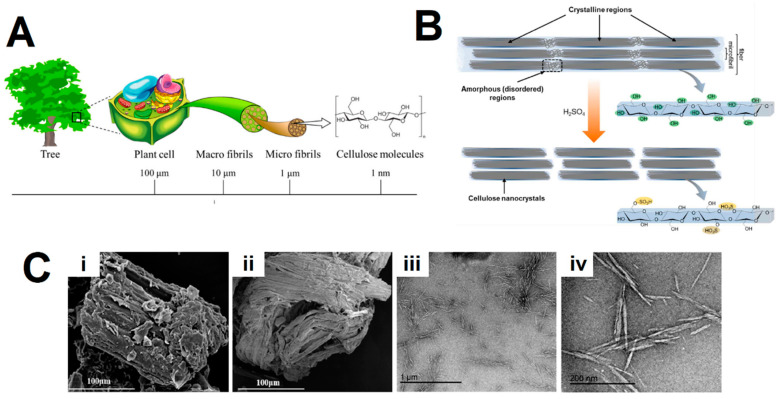
(**A**) Cellulose contained in plants or trees has a hierarchical structure from the meter to the nanometer scale. Adapted from reference [5]. (**B**) Schematics of idealized cellulose fibers showing one of the suggested configurations of the crystalline and amorphous regions, and CNCs after sulfuric acid hydrolysis of the amorphous regions, exhibiting the characteristic sulfate half-ester surface groups formed as a side reaction. Adapted from reference [6]. (**C**) Scanning electron microscope (SEM) images of (i) the raw corn stalk, (ii) the extracted cellulose and (iii,iv) transmission electron microscopy (TEM) images of the isolated cellulose nanocrystals (CNCs) using sulfuric acid hydrolysis. Adapted from reference [7].

**Figure 2 gels-09-00574-f002:**
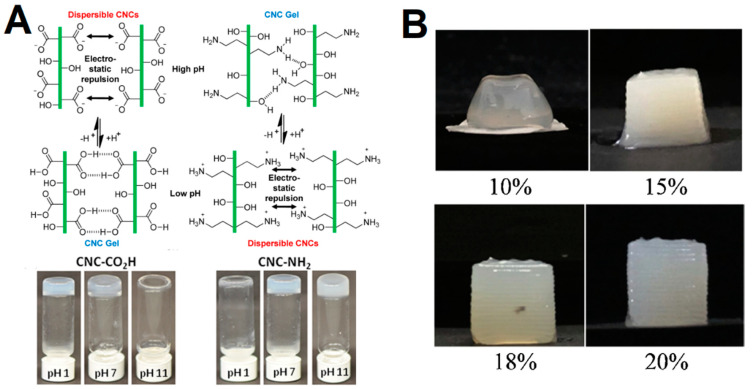
(**A**) Schematic representation of the proposed interactions between CNC-CO_2_H and CNC-NH_2_ at high and low pH; Images of aqueous dispersions of 2.7 wt% CNC-CO_2_H and 2.7 wt% CNC-NH_2_ at pH 1, 7, and 11. Adapted from reference [28]. (**B**) Visual appearance of 1 cm^3^ cubes printed with CNC hydrogels of increasing solid loading. Adapted from reference [29].

**Figure 3 gels-09-00574-f003:**
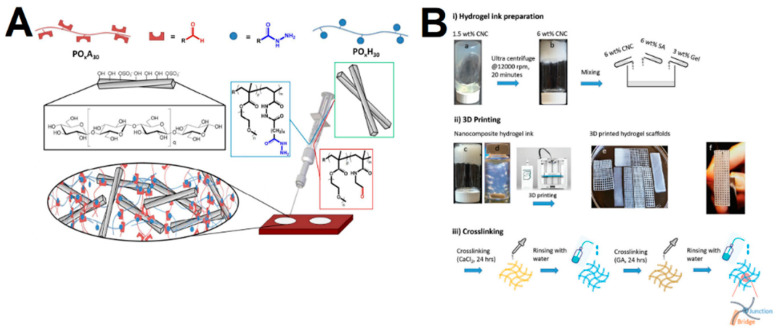
(**A**) Schematic representation of injectable CNC-reinforced poly(oligoethylene glycol methacrylate) (POEGMA) gels. Adapted from reference [74]. (**B**) Schematic representation of the processing route for 3D-printed nanocomposites hydrogel scaffolds of sodium alginate (SA) and gelatin (G) reinforced with cellulose nanocrystals (CNCs). Adapted from reference [78].

**Figure 4 gels-09-00574-f004:**
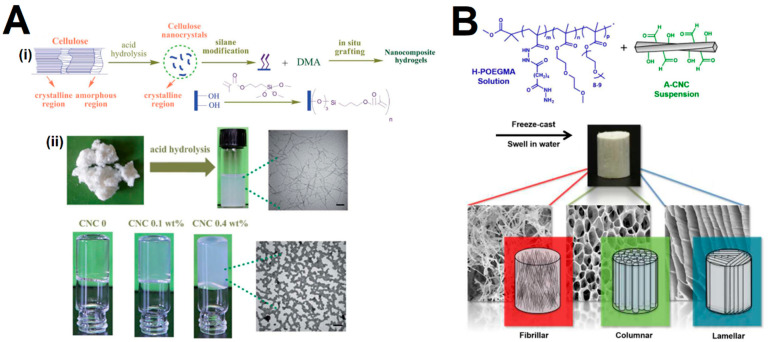
(**A**) (i) Schematic representation of the preparation of silane-modified CNCs to fabricate nanocomposite hydrogels, and (ii) TEM images (bar = 100 nm) of the CNCs after acid hydrolysis and the percolating gels obtained from free radical polymerization of poly(N,N-dimethylacrylamide) (DMA). Adapted from reference [98]. (**B**) Schematic representation of the fabrication of CNC-POEGMA nanocomposite hydrogels formed through hydrazone cross-linking of aldehyde-modified CNCs (A-CNCs) with hydrazide-functionalized POEGMA (H-POEGMA) and subsequent directional freeze casting, and the SEM images of the resulting hydrogels’ morphologies (fibrillar, columnar, lamellar). Adapted from reference [94].

**Table 2 gels-09-00574-t002:** Studies on CNC composite hydrogel applications.

Components	Notes	Cargo	Application	Ref.
CNCs; 5-aminolevulinic acid (ALA); dopamine (DPA)	Increased adhesion of the nanohydrogels to cells, and reactive oxygen species (ROS) production	Paclitaxel (PTX)	Cancer therapy	[67]
poly(ε-caprolactone-*co*-lactide)-*b*-poly(ethylene glycol)-*b*-poly(ε-caprolactone-*co*-lactide (PCLA); CNCs	Good injectability and high shape fidelity in 3D printing	DOX	Cancer therapy	[150]
Xanthan (XG); Chitosan (CS); CNCs	Improved mechanical performance	5-Flurouracil	Tissue engineering Drug delivery	[151]
hyaluronic acid (HA); CNCs; Enriched with platelet lysate	Enhanced cells’ viability and angiogenic activity	Chemotactic and pro-angiogenic growth factors	Tissue regeneration	[152]
Chitosan-ulvan hydrogel; CNCs	Fast wound-healing efficiency	Epidermal growth factor (EGF)	Wound healing	[153]
CNCs decorated with Fe_3_O_4_ nanoparticles; poly(N-isopropylacrylamide) (PNIPAm)	High drug-loading content (10.18 g/g) Thermo-response triggered by NIR	Vancomycin	Wound healing	[65]
CS; CNCs	Hydrogel scaffold degrades according to a preferred route	Bovine serum albumin (BSA)	Release of macromolecules	[154]
CNC gels formed by salt-induced charge screening	Drug release modulated by the incorporation of sucrose or xanthan gum	BSA; Tetracycline (TC); Doxorubicin (DOX)	Drug delivery	[155]
CS; CNCs	Improved cell viability and mineralization Enhanced osteogenic-related gene expression Enhanced antibacterial activity	TC	Osteogenesis; Antibacterial agent; Drug delivery	[156]
Magnetic nanocellulose (m-CNCs) alginate hydrogel beads	m-CNCs increased the integrity and the swelling percentage	Ibuprofen	Drug delivery	[157]

## Data Availability

Not applicable.

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
