# Peer review of "Cellulose Nanocrystal (CNC) Gels: A Review"

_gels, 2023, doi:10.3390/gels9070574_

Round 1

Reviewer 1 Report

Dear authors,

The manuscript represent and discuss a good results and previous works,

I just recommand to support the introduction with more references.

Best regards.

Reviewer 2 Report

This is a nice review in a very actual field of polymer science. Since cellulose is a natural and biodegradble polymer with many (potential) applications it fits well in the actual discussion of 'plastics' in the environment. The review contains many citations also most recent. The graphics are well prepared and several topics are summarized in Tables. So I highly recommend publication of the review in its current form. A minor point is that the authors should also cite the book by Wadwood, Hamad 'Cellulose Nanocrystals'. In the introdution the authors focus on sulfuric acid. But also other systems are equally impostant as nitric acid/ phosphoric acid.

Reviewer 3 Report

This manuscript can be accepted for publication after the authors provide sufficient response to the following comments: comparison with current reviews in CNC should be provided 
